# FAST OBJECT LOCALIZATION VIA SENSITIVITY ANALYSIS

## ABSTRACT

Deep Convolutional Neural Networks (CNNs) have been repeatedly shown to perform well on image classification tasks, successfully recognizing a broad array of objects when given sufficient training data. Methods for object localization, however, are still in need of substantial improvement. In this paper, we offer a fundamentally different approach to the localization of recognized objects in images. Our method is predicated on the idea that a deep CNN capable of recognizing an object must implicitly contain knowledge about object location in its connection weights. We provide a simple method to interpret classifier weights in the context of individual classified images. This method involves the calculation of the derivative of network generated activation patterns, such as the activation of output class label units, with regard to each input pixel, performing a sensitivity analysis that identifies the pixels that, in a local sense, have the greatest influence on internal representations and object recognition. These derivatives can be efficiently computed using a single backward pass through the deep CNN classifier, producing a *sensitivity map* of the image. We demonstrate that a simple linear mapping can be learned from sensitivity maps to bounding box coordinates, localizing the recognized object. Our experimental results, using real-world data sets for which ground truth localization information is known, reveal competitive accuracy from our fast technique.

## 1 INTRODUCTION

Deep Convolutional Neural Networks (CNNs) have been shown to be effective at image classification, accurately performing object recognition even with thousands of object classes when trained on a sufficiently rich data set of labeled images Krizhevsky et al. (2012). One advantage of CNNs is their ability to learn complete functional mappings from image pixels to object categories, without any need for the extraction of hand-engineered image features Sermanet et al. (2013). To facilitate learning through stochastic gradient descent, CNNs are (at least approximately) differentiable with regard to connection weight parameters.

Image classification, however, is only one of the problems of computer vision. In the task of image classification, each image has a single label, associated with the class identity of the main object in the image, and the goal is to assign correct labels in a manner that generalizes to novel images. This can be accomplished by training a machine learning classifier, such as a CNN, on a large data set of labeled images Deng et al. (2009). In the object localization task, in comparison, the output for a given image is not a class label but the locations of a specified number of objects in the image, usually encoded as bounding boxes. Evaluation of an object localization system generally requires ground truth bounding boxes to compare to the system's output. The detection task is more difficult than the localization task, as the number of objects are not predetermined Sermanet et al. (2013).

In this paper, we focus on object localization, identifying the position in the image of a recognized object. As is common in the localization literature, position information is output in the form of a bounding box. Previously developed techniques for accomplishing this task generally involve searching the image for the object, considering many candidate bounding boxes with different sizes and locations, sometimes guided by an auxilliary algorithm for heuristically identifying regions of interest Sermanet et al. (2013); Girshick (2015); He et al. (2017). For each candidate location, the sub-image captured by the bounding box is classified for object category, with the final output

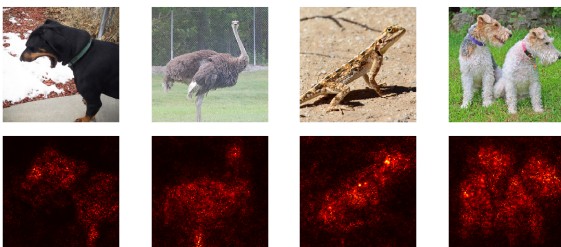

Figure 1: Examples of sensitivity maps, displaying the sensitivity of network internal representations to individual pixels, providing information about the locations of the main objects in the source images.

bounding box either being the specific candidate region classified as the target object with the highest level of certainty or some heuristic combination of neighboring or overlapping candidate regions with high classification certainty. These approaches tend to be time consuming, often requiring deep CNN classification calculations of many candidate regions at multiple scales. Efforts to speed these methods mostly focus on reducing the number of regions considered, typically by using some adjunct heuristic region proposal algorithm Girshick (2015); Ren et al. (2015); He et al. (2017). Still, the number of considered regions is often reported to be roughly 2,000 per image. While these approaches can be fairly accurate, their slowness limits their usefulness, particularly for online applications.

A noteworthy alternative approach is to directly train a deep CNN to produce outputs that match ground truth localization bounding boxes, using a large image data set that provides both category and localization information for each image. It appears as if some form of this method was used with AlexNet Krizhevsky et al. (2012), though details concerning localization, rather than image classification, are difficult to discern from the published literature. A natural approach would be to cast the learning of bounding boxes as a simple regression problem, with targets being the four coordinates that specify a bounding box (e.g., coordinates of upper-left and lower-right corners, or region center coordinates along with region width and height). It is reasonable to consider sharing early layers of a deep CNN, such as those performing convolution and max pooling, between both an image classification network and an object localization network. Indeed, taking such a multitask learning approach Caruana (1997) can allow for both object category and object location training data to shape connection weights throughout the network. Thus, the deep CNN would have "two heads", one for image classification, using a classification cross-entropy loss function, and one for object localization, reducing the $\ell_2$ norm between ground truth and predicted bounding box coordinates Krizhevsky et al. (2012). While this approach can produce a network that quickly outputs location information, extensive training on large data sets containing ground truth bounding box information is necessary to produce good generalization.

In this paper, we introduce an approach to object localization that is both very fast and robust in the face of limited ground truth bounding box training data. This approach is rooted in the assertion that any deep CNN for image classification must contain, implicit in its connection weights, knowledge about the location of recognized objects Selvaraju et al. (2016). The goal, then, is to interpret the flow of activation in an object recognition network when it is performing image classification so as to extract information about object location. Furthermore, the goal is to do this quickly. Thus, this approach aims to leverage location knowledge that is already latent in extensively trained and tuned image classification networks, without requiring a separate learning process for localization.

Our method makes use of the notion of a *sensitivity analysis* Sobol (1993). We propose estimating the sensitivity of the category outputs, or activation patterns at internal network layers, of an image classification CNN to variance in each input pixel, given a specific input image. The result is a numeric value for each pixel in the input image that captures the degree to which small changes in that pixel (locally, around its current value) give rise to large changes in the output category. Together, these numeric values form a *sensitivity map* of the image, encoding image regions that are important for the current classification. Our proposed measure of sensitivity is the partial derivative of activity with regard to each pixel value, evaluated for the current image. For a deep CNN that formally embodies a differentiable mapping (at least approximately) from image pixels to output categories, this partial derivative can be quickly calculated. While many tools currently exist for

efficiently calculating such derivatives, we provide a simple algorithm that computes these values through a single backward pass through the image classification network, similar to that used to calculate unit error (delta) values in the *backpropagation of error* learning algorithm Rumelhart et al. (1986). Thus, we can generate a sensitivity map for an image in about the same amount of time as it takes the employed image classification network to produce an output. Some example sensitivity maps are shown in Figure 1.

The idea of using sensitivity information, like that in our sensitivity maps, for a variety of tasks, including localization, has previously appeared in the literature Simonyan et al. (2013); Zhou et al. (2016); Selvaraju et al. (2016). Indeed, some of these past efforts have used more sophisticated measures of sensitivity. In this paper, we show that even our very simple sensitivity measure can produce strong localization performance, and it can do so quickly, without any modifications to the classification network, and even for object categories on which the classification network was not trained. The relationship of the results reported here to previously reported work is discussed further in Section 4.

As previously mentioned, object localization methods typically encode object location as a bounding box. Since our sensitivity maps encode location differently, in terms of pixels, we propose learning a simple linear mapping from sensitivity maps to bounding box coordinates, allowing our method to output a bounding box for each classified image. We suggest that this linear mapping can be robustly learned from a relatively small training set of images with ground truth bounding boxes, since the sensitivity maps form a much more simple input than the original images.

The primary contributions of this paper may be summarized as follows:

- We propose a new general approach to performing object localization, interpreting a previously trained image classification network by performing a sensitivity analysis, identifying pixels to which the category output, or a more general internal representation, is particularly sensitive.

- We demonstrate how a linear function from the resulting sensitivity maps to object location bounding box coordinates may be learned from training images containing ground truth location information.

- We provide a preliminary assessment of our approach, measuring object localization performance on the ImageNet and PASCAL VOC data sets using the VGG16 image classification CNN, showing strong accuracy while maintaining short computation times.

## 2 METHOD

### 2.1 CALCULATING PIXEL SENSITIVITIES IN A TRAINED CNN

Calculating derivatives of a function of network output with regard to network parameters, such as connection weights, is a standard part of CNN training. It is common for learning in a deep CNN to involve stochastic gradient decent, which involves such derivatives. In that case, the derivatives are of an objective function with regard to connection weight values. In image classification networks, the objective function is designed to have optima where training images are correctly classified. In the case of object localization, a similar objective function could be designed to minimize differences between output bounding box coordinates and provided ground truth bounding box coordinates, for all images in an appropriately labeled training set. For example, given $N$ training images, stored in the matrix $\mathbf{X}$, with the ground truth 4-dimensional bounding box vector for image $x_i$ being $y_i$, and $G(x_i; \mathbf{w})$ being the CNN output vector for image $x_i$ given connection weights $\mathbf{w}$, an appropriate loss function would be:

$$\ell(\mathbf{X}, \mathbf{w}) = \frac{1}{N} \sum_{i=1}^{N} \|y_i - G(x_i; \mathbf{w})\|_2^2 \qquad (1)$$

The CNN will produce good estimates of the training image bounding boxes when this loss function is minimized with regard to $\mathbf{w}$. Network weight parameters that minimize this loss, $\mathbf{w}^*$, may be sought through stochastic gradient decent, incrementally updating $\mathbf{w}$ according to the gradient of $\ell(\mathbf{X}, \mathbf{w})$ with regard to $\mathbf{w}$. A primary drawback of this approach is that it requires a large and representative sample of images with ground truth bounding box information.

Consider that, once weights are found, the gradient of $\ell(\mathbf{X}, \mathbf{w}^*)$ with regard to $\mathbf{X}$ would provide information about the sensitivity of the bounding box loss function with regard to the pixels in the images. This gradient can be calculated as efficiently as the gradient of the loss with regard to the weights, with both depending on the gradient of $G(x_i; \mathbf{w})$ with regard to a subset of its arguments. This means that the gradient of $G(x_i; \mathbf{w}^*)$ with regard to $x_i$ can be efficiently computed, and that gradient would capture the sensitivity of bounding box coordinates with regard to the specific pixels in image $x_i$. Note that this gradient can be calculated for images beyond those in the training set. Knowing which pixels in a novel image play an important role in determining the bounding box provides useful information for object localization. Using this calculation to address the object localization task makes little sense, however, as $G(x_i; \mathbf{w}^*)$ provides an estimate of object location without a need to consider pixel sensitivity.

Rather than training a deep CNN to output bounding boxes, requiring extensive labeled data, we propose calculating the same gradient for a different network – one successfully trained to perform image classification. If we now see $G(x_i; \mathbf{w}^*)$ as the output of such an image classification network, its gradient with regard to $x_i$ would provide information about the sensitivity of the assigned category to individual pixels. Pixels with the largest absolute values of this derivative will, around the input $x_i$, produce the largest changes in the classification decision of the CNN. This can be seen as one measure of how important pixels are for classifying the object in the image. Consider that the object class output is not immediately affected by changes to pixels with a derivative of zero.

The calculation of this gradient can be performed as efficiently as a single "backward pass" through the classification network. This is well illustrated by considering the case of a simple layered back-propagation network Rumelhart et al. (1986) in which the "net input" of unit $i$, $\eta_i$, is a weighted sum of the activations of units in the previous layer, and the activation of unit $i$ is $g(\eta_i)$, where $g(\cdot)$ is the unit activation function. In this case, we can define a sensitivity value for each unit, $s_i$, as the derivative of the network output with regard to $\eta_i$. Using the chain rule of calculus, it is easy to show that the sensitivity of an output unit is $g'(\eta i)$, and, for units in earlier layers the gradients are computed as follows:

$$s_i \;=\; g'(\eta_i) \sum_k w_{ki}\, s_k \tag{2}$$

where $k$ iterates over all units in the immediately downstream layer from unit $i$ and $\mathbf{w}_{ki}$ is the connection weight from unit $i$ to unit $k$. This calculation may be performed, layer by layer, from outputs to inputs, until $s_i$ values for each pixel input unit are available.

This demonstrates how efficiently pixel sensitivity values can be calculated for a given classified image. Of course, there are currently a variety of software packages that include tools for calculating gradients. In the evaluation of our approach in Section 3, we report results using the tools provided by TensorFlow Abadi et al. (2015).

## 2.2 Sensitivity of the Attention Map

We have proposed using a previously trained image classification network as a source of information about object location, focusing on the gradient of the network output with regard to image pixels. It is interesting to note that it might not be necessary to perform the sensitivity calculation using the full classification network. There is a growing body of research that suggests that, in a well trained image classification CNN, the features that are extracted at the "attention map" layer (i.e., the output of the last convolutional layer) tend to be generally useful for learning a variety of image analysis tasks Razavian et al. (2014); Donahue et al. (2014). Inspired by these results, we have investigated the possibility of substituting the gradient of the classifier output with regard to pixels with the gradient of the attention map with regard to pixels. This avoids calculations involving final fully connected layers and any classification softmax layer. Generating image sensitivity maps from the attention map layer is slightly faster than our original proposal, but, more importantly, it is possible that general knowledge about object location might be found in the attention map, and using the attention map as the basis of the sensitivity map might actually generalize beyond the categories on which the image classiciation CNN was trained. We have not yet done a formal comparison of these two approaches to constructing the sensitivity map, but example results using both approaches are reported in Section 3. Note that computing the gradients of the aggregated values of the last convolution layer with respect to the input pixels are considered as the Gestalt total which can be

computed as follows.

$$GT = \frac{1}{H \times W \times C} \sum_{i,j,k} A_n(i,j,k) \tag{3}$$

which $A_n$ is the activation map of the last convolution layer, $H, W$ and $C$ are the height, width and channels of the last convolution layer; moreover, $GT$ indicates as Gestalt Total.

### 2.3 Aggregating Across Color Channels

The sensitivity map calculations that have been described, so far, provide a scalar sensitivity value for each *input* to the image classification deep CNN. Color images, however, are regularly provided to such networks using multiple inputs per image pixel, often encoding each pixel over three color channels. Thus, the gradient calculation will actually produce three sensitivity values for each pixel. Since we hope to produce a sensitivity map that focuses in a general way on location information, it seems reasonable to aggregate the three sensitivity values into one. Since the direction of the sensitivity relationship with the class output is irrelevant, a good first step is to take the absolute value of each derivative. Given that dependence on even a single color channel suggests that a pixel is important for identifying the object, an argument can be made that a pixel should be labeled with the maximum of the three absolute derivatives. Alternatively, it could be argued that all color channels should be taken into account when producing the sensitivity map, in which case it might be better to average the three absolute derivatives. We have explored both of these aggregation methods, with results appearing in Section 3.

### 2.4 Learning to Produce Bounding Boxes

Object localization algorithms typically output the four coordinates of a bounding box to communicate the location of the target object. Such a bounding box is not intrinsic to a sensitivity map, however. Heuristic techniques could be used to identify a rectangular region that captures the majority of the high sensitivity pixels, while avoiding low sensitivity pixels, but we have taken a different approach. We have opted to learn a linear mapping from sensitivity maps to bounding box coordinates, using training images with ground truth location information.

It is important to note that learning this mapping is not the same as learning to map from the original images to bounding box coordinates, as has been done in some other object localization systems. Sensitivity maps contain much less information than the original images, so using the sensitivity maps as inputs both reduces the dimensionality of the input to this mapping and makes for a more simple functional relationship between pixels and bounding box coordinates. We expect that this simplification will allow the mapping to bounding box coordinates to be successfully learned using a far smaller set of training images labeled with ground truth object locations. Indeed, we expect that a simple linear mapping could perform well.

Formally, we define the parameters of the linear mapping to the four bounding box coordinates as a $4 \times M$ matrix, $\hat{W}$, (where $M$ is the number of pixels in an image) and a 4-dimensional vector of "bias weights", $\hat{w}$. Given a sensitivity map, $s$, the output is $(\hat{W}s + \hat{w})$. Given a training set of $N$ images, the mapping is found by minimizing the following objective function with regard to $\hat{W}$ and $\hat{w}$:

$$\frac{1}{N} \sum_{i=1}^{N} \frac{1}{4} \sum_{j=1}^{4} \|B_{i,j} - (\hat{W}s_i + \hat{w})\|_2^2 \tag{4}$$

where $s_i$ is the sensitivity map for the $i^{th}$ image, and $B_{i,j}$ is the $j^{th}$ coordinate of the bounding box for the $i^{th}$ image. This learning process amounts to four independent linear regression problems, which can be solved efficiently.

Once learned, mapping from sensitivity maps to bounding box coordinates can be done very quickly. With sensitivity map formation requiring only a single backward pass through the image classification network, the whole process – from image, to classification, to sensitivity map, to bounding box – can be performed in little more than twice the time it takes for the network to do object recognition.

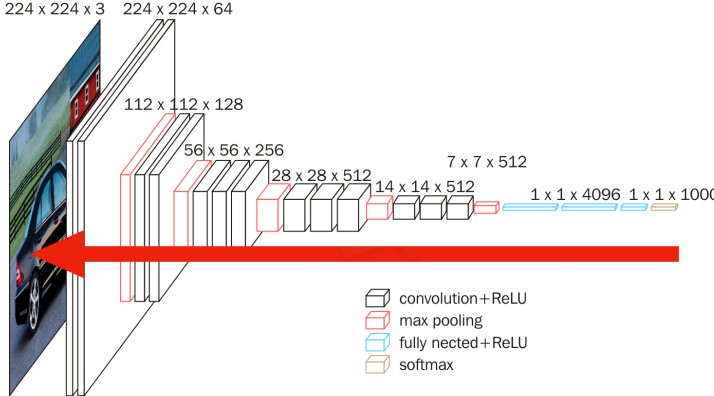

Figure 2: A schematic illustration of the proposed method for sensitivity analysis on the pre-trained VGG16 network.

## 3 RESULTS

The code and the sensitivity map of on imageNet as well as PASCAL VOC dataset will be publicly available.

### 3.1 DATA SETS & PERFORMANCE MEASURES

We evaluated our proposed method for object localization on two challenging data sets: the PASCAL VOC 2007 Everingham et al. (2008) data set and the ImageNet 2012 Deng et al. (2009) data set. The PASCAL VOC 2007 data set was selected due to its use in the existing object localization literature. The ImageNet data set is one of the largest publicly available data sets. It also contains many images annotated with ground truth bounding boxes.

We followed the literature with regard to the evaluation criterion applied to our method, using *Cor-Loc*, which has been used for weakly supervised localization. The CorLoc metric is defined as the percentage of images in a data set that are correctly localized based on the PASCAL criterion, in which a given localization is considered correct if and only if the *intersection over union (IOU)* area of the predicted and ground truth bounding boxes is greater than one half:

$$IOU = \frac{area(\beta_p \cap \beta_{gt})}{area(\beta_p \cup \beta_{gt})} > 0.5 \qquad (5)$$

...where $\beta_p$ is the predicted bounding box and $\beta_{gt}$ is the ground truth bounding box Tang et al. (2014).

### 3.2 PRE-TRAINED IMAGE CLASSIFICATION DEEP CNN

To demonstrate that our approach works with an image classification deep CNN that was in no way specialized for our localization method, we opted to use a publicly available network. We used the VGG16 network, shown in Figure 2, fully trained Simonyan & Zisserman (2014). This network provides ImageNet object classes as output, allowing us to calculate sensitivity maps based on the network classification when examining ImageNet data. For the PASCAL VOC 2007 data set, we used the previously described method of calculating derivatives based on the attention map of VGG16, since there is not consistent class correspondnce between the PASCAL VOC 2007 classes and the classes on which VGG16 was trained. To produce sensitivity maps for the PASCAL VOC 2007 data set, we aggregated across color channels by using the maximum absolute derivative across the three inputs for each pixel. For the ImageNet data set, we averaged the absolute derivatives across the three inputs in order to produce pixel sensitivity values.

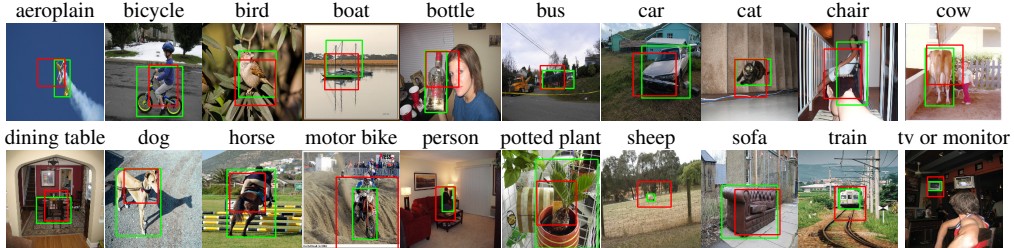

Figure 3: Results of the proposed method on the first 10 classes of PASCAL VOC 2007. Each column shows three examples in one class. The green boxes are the ground truth, and the red ones are the predicted bounding boxes.

## 3.3 EXPERIMENTAL SETUP

For generating sensitivity maps, we have used a pretrained vgg 16 network, we have used the whole network architecture while we were experimenting on ImageNet dataset, otherwise we have removed the last 3 fully connected layers and computed the Gestalt Total by the last convolution layer. The derivatives in either case computed using just one backward pass to the original pixels. For learning bounding boxes we have used the aggregated sensitivity maps as an input. To learn the mapping from sensitivity maps to bounding box coordinates, we performed linear regression using stochastic gradient decent. Updates were performed in batches of 2,048. The learning rate was initialized to 0.1 and decayed by a factor of 10 every 10,000 iterations. The experiment run on 1 GPU for 4 days.

## 3.4 PERFORMANCE ON PASCAL VOC 2007

The full PASCAL VOC 2007 data set includes 12,608 training set images and an equal number of testing set images Everingham et al. (2008). Each image contains an object of 1 of 20 different categories. We applied our object localization method to this full data set. However, we were unable to find published localization performance data for other methods applied to the full data set, which we might use for comparison to our approach. Work reported in Tang et al. Tang et al. (2014) provides performance data on 6 of the classes: aeroplane, bicycle, boat, bus, horse, and motorbike. Performance on these same classes have also been reported by others Russell et al. (2006); Chum & Zisserman (2007); Deselaers et al. (2012). Table 1 compares the localization performance of our method with that of other approaches. Note that our method, while being very fast, outperforms the comparison algorithms.

Table 1: CorLoc Performance on PASCAL VOC 2007 (6 Classes)

| Method | Average CorLoc |
|---|---|
| Russell et al. | 22% |
| Chum & Zisserman | 32% |
| Deselaers et al. | 37% |
| Tang et al. | 39% |
| **Sensitivity Maps** | **55%** |

Examples of the bounding boxes selected by our method, compared to ground truth, for all 20 classes in the PASCAL VOC 2007 data set are shown in Figure 3. Qualitatively, it appears as if our approach is most accurate when there is a single target object with little crowding. However, if the target object is small and in a crowded region of the image, performance is less reliable.

While speed is an important property of our method, as is the reuse of classification training for localization, we compared our approach to data from some slower state-of-the-art deep learning techniques for localization that do not necessarily have these properties. We compared our method to R-CNN Girshick et al. (2014), DPM Felzenszwalb et al. (2013), and Poselets Bourdev & Malik (2009). These were chosen due to the ready availability of published localization results for these alternative methods on the PASCAL VOC 2007 data set, with the measure of performance being Average CorLoc (or mean Average Precision, mAP). The comparison results are given in Table **??**. Several of the comparison methods display better localization performance than our approach, but

Table 2: PASCAL VOC 2007 Test Detection Results. The proposed method performed favorably against the state of the art methods.

| Method | MeanCL | areo | bike | bird | boat | bottle | bus | car | cat | chair | cow | table | dog | horse | mbike | person | plant | sheep | sofa | train | tv |
|---|---|---|---|---|---|---|---|---|---|---|---|---|---|---|---|---|---|---|---|---|---|
| Siva et al. (2012) | 30.2 | 45.8 | 21.8 | 30.9 | 20.4 | 5.3 | 37.6 | 40.8 | 51.6 | 7.0 | 29.8 | 27.5 | 41.3 | 41.8 | 47.3 | 24.1 | 12.2 | 28.1 | 32.8 | 48.7 | 9.4 |
| Shi et al. (2013) | 36.2 | 67.3 | 54.4 | 34.3 | 17.8 | 1.3 | 46.6 | 60.7 | 68.9 | 2.5 | 32.4 | 16.2 | 58.9 | 51.5 | 64.6 | 18.2 | 3.1 | 20.9 | 34.7 | 63.4 | 5.9 |
| Gokberk Cinbis et al. (2014) | 38.8 | 56.6 | 58.3 | 28.4 | 20.7 | 6.8 | 54.9 | 69.1 | 20.8 | 9.2 | 50.5 | 10.2 | 29.0 | 58.0 | 64.9 | 36.7 | 18.7 | 56.5 | 13.2 | 54.9 | 59.4 |
| OM Li et al. (2016) | 31.8 | 50.4 | 30 | 34.6 | 18.2 | 6.2 | 39.3 | 42.2 | 57.3 | 10.8 | 29.8 | 20.5 | 41.8 | 43.2 | 51.8 | 24.7 | 20.8 | 29.2 | 26.6 | 45.6 | 12.5 |
| SP-VGGNet Zhu et al. | 60.6 | 85.3 | 64.2 | 67.0 | 42.0 | 16.4 | 71.0 | 64.7 | 88.7 | 20.7 | 63.8 | 58.0 | 84.1 | 84.7 | 80.0 | 60.0 | 29.4 | 56.3 | 68.1 | 77.4 | 30.5 |
| PBRM Cho et al. (2015) | 36.6 | 50.3 | 42.8 | 30.0 | 18.5 | 4.0 | 62.3 | 64.5 | 42.5 | 8.6 | 49.0 | 12.2 | 44.0 | 64.1 | 57.2 | 15.3 | 9.4 | 30.9 | 34.0 | 61.6 | 31.5 |
| Sensitivity Maps | **40.1** | 63.8 | 55.1 | 41.2 | 23.3 | 34.2 | 58.6 | 72.7 | 36.9 | 23.3 | 49.7 | 11.5 | 29.6 | 50.1 | 65.9 | 11.8 | 42.2 | 39.7 | 18.1 | 51.0 | 41.2 |

it is important to keep in mind that the comparison cases had some important advantages, including taking the time to use a sliding window and access to the class labels on which the network was trained. Recall that our sensitivity maps were produced, in this case, by calculating the sensitivity of the network attention map activity to pixel values. Thus, this comparison illustrates trade-offs between speed, performance, and generalization.

Note that as one of the reviewers mentioned it would be worth looking at the results if we just use the sensitivity maps and heuristics to draw bounding boxes out of objects. For this experiment, we used a Gaussian smoothing filter to smooth out the sensitivity maps and then we picked top %20 pixels and draw the bounding box out of those pixels as other researchers obtained this experiment before Zhou et al. (2016); Selvaraju et al. (2016). Based on our observation we noticed that it could damage the mean CorLoc by %3 in our best observations. However, this process is highly depends on the smoothing $\sigma$ parameter. The obtained results from different $\sigma$ values are reported in Table 3.

Table 3: Average CorLoc Performance on Pascal VOC 2007 based on heuristic bounding box

| $\sigma$ | CorLoc |
|---|---|
| 10 | 27.2% |
| 20 | 38.4% |
| 30 | 32.5% |

## 3.5 PERFORMANCE ON IMAGENET

ImageNet is a large image data set that has been systematically organized by object category Deng et al. (2009). We executed a large scale evaluation of our approach by using all images in ImageNet that are annotated with ground truth localization information. This subset contains 300,916 images involving 478 object classes. We divided this data set into a training set, a test set, and a validation set by sampling without replacement (i.e., the intersection between each pair of the three sets was empty). There were 225,687 images (75%) in the training set, and there were 45,137 images in each of the other two sets.

We compared the performance of our approach with two methods discussed in Tang et al. Tang et al. (2014) for which ImageNet results are explicitly reported: Top Objectiveness Box & Co-Localization. Also, we noted that many images in this data set presented the target object in the middle of the image, providing a bias that could be leveraged by learned localization systems. Thus, as a baseline of performance, we calculated the CorLoc performance for a system that blindly offered the same bounding box in the middle of the image, with average size, for every input. The results are shown in Table 4. Once again, note the relatively high accuracy performance of our efficient method. Also note that the baseline was comfortingly low. As might be expected, performance varies with

Table 4: CorLoc Performance on ImageNet (478 Classes)

| Method | Average CorLoc |
|---|---|
| Constant Center Box Baseline | 12.34% |
| Top Objectiveness Box | 37.42% |
| Co-Localization | 53.20% |
| **Sensitivity Maps** | **68.76%** |

class. Our algorithm appears to do well on some objects, such as balls and dogs. One might suspect

that failures arise in the linear mapping from sensitivity maps to bounding box coordinates, but a perusal of the sensitivity maps, themselves, suggests that the pixel sensitivity values vary in utility across different object categories. Still, our method performs fairly well across the classes. Note that the IOU does not fall below 0.62 for any class. This suggests that, while some individual images may be problematic, the overall performance for each class is quite good. This universally strong class-specific performance is also displayed in Table 4.

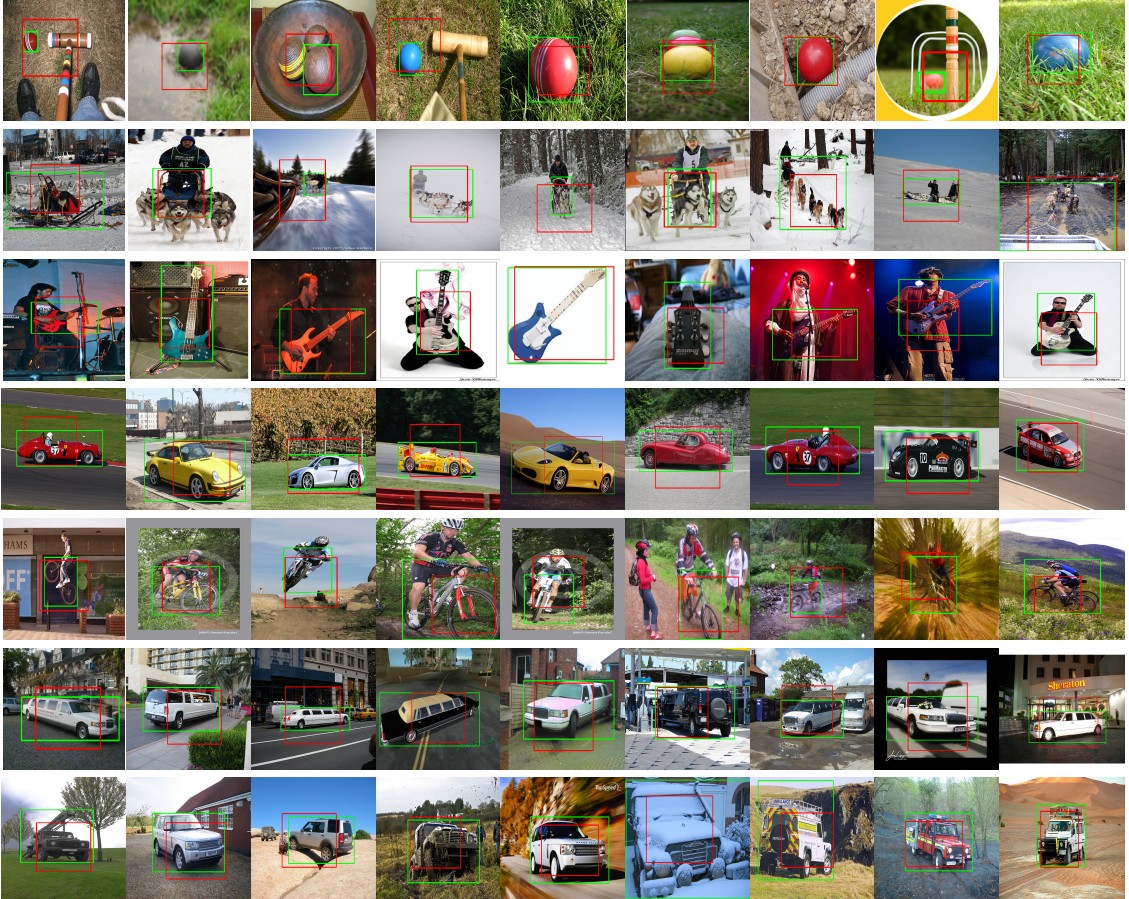

Figure 4: Results of the proposed method on different object categories from the ImageNet data set. Each row shows 9 examples in one class. The green boxes are the ground truth, and the red ones are the predicted bounding boxes.

## 3.6 AGGREGATION METHODS ANALYSIS

The sensitivity analysis approach gives us the sensitivity of every single pixels in all channels in the RGB images and since we are in need of locations we need to aggregate among channels. We proposed two methods, an average function, a maximum function. The first approach is to taking average among channels and the second method is to pick up the maximum numbers among channels. We didn't notice significant difference between these two methods in the localization performance, the only information that comes to light is generating sensitivity maps based on average function is a bit smoother in visual sense than maximum function. The CorLoc between average and maximum aggregation function on ImageNet dataset are 68.7 and 67.9 respectively and the results of these two aggregation operators on PASCAL VOC dataset is 39.2 and 40.1 ,respectively.

### 3.7 SPEED ANALYSIS

To analyze our object localization approach since it highly depends to the hardware parameters and network architecture, we decided to analyze the speed in term of forward and backward passes. Our approach only needs two passes, one forward pass for the classification and one backward pass for localization. If we consider each forward/backward pass as $n$ operations we can say our approach is $O(N^2) + \epsilon$ which means it needs one forward pass, one backward pass and one inference from the linear model.

## 4 CONCLUSION

We have presented an approach to object localization based on performing a sensitivity analysis of a previously trained image classification deep CNN. Our method is fast enough to be used in online applications, and it demonstrates accuracy that is superior to some methods that are much slower. It is likely that even better accuracy could be had by incorporating sensitivity analysis information into a more sophisticated bounding box estimator.

As previously noted, the idea of using sensitivity information has appeared in previously published work. There are ways in which the results reported in this paper are distinct, however. We have moved beyond visualization of network function using sensitivity (or *saliency*) Simonyan et al. (2013) to performing direct comparisons between different methods on the localization task. We have shown that using a fast and simple measure of sensitivity can produce comparable performance to that of much slower methods. Our approach produces good generalization without modifying the classification network, as is done in *Class Activation Mapping (CAM)* Zhou et al. (2016). With our PASCAL VOC 2007 results, we have shown that our approach can successfully be applied to attention maps, even when the image contains objects belonging to a class on which the classification network was not trained, distinguishing it from *Grad-CAM* Selvaraju et al. (2016). In short, we have demonstrated the power of a simple sensitivity measure for performing localization.

Note that our approach may be used with image classifiers other than CNNs. The proposed sensitivity analysis can be conducted on any differentiable classifier, though performance will likely depend on classifer specifics. Indeed, at a substantial time cost, even a black box classifier could be approximately analyzed by making small changes to pixels and observing the effects on activation patterns.

The proposed approach is quite general. Indeed, we are currently working on applying sensitivity analysis to deep networks trained on other tasks, with the goal of interpreting network performance on the current input in a useful way. Thus, we see a potentially large range of uses for sensitivity analysis in neural network applications.

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
