# OpenReview forum: "FAST OBJECT LOCALIZATION VIA SENSITIVITY ANALYSIS"
_ICLR.cc/2019/Conference_

### Official Review · AnonReviewer3 · 2018-10-29
**Poor readibility, lack of insight and fundamental background knowledge, reject**

**Rating:** 3
**Confidence:** 4

**Review:**

summary--
The paper focuses on improving object localization, though the title highlights "interpreting deep neural network" which is another area. It analyzes the classifier weights for image classification, and compute the derivative of the feature maps from the network for a sensitivity map of the image. Then it learns a simple linear mapping over the sensitivity map for bounding box regression. Experiments report competitive performance.

However, there are several major concerns.

1) The paper appears misleading from multiple claims. For example, [abstract] "common approaches to this problem involve the use of a sliding window,... time consuming". However, current state-of-the-art methods accomplish detection in a fully convolutional manner using CNN, and real-time performance is achieved. the paper claims that "computer vision can be characterized as presenting three main tasks... (1) image classification, (2) image localization and (3) image detection". This appears quite misleading. There are way more topics, from low-level vision to mid-level to high-level, e.g., stereo, boundary detection, optical flow, tracking, grouping, etc. Moreover, just in "localization", this could be object localization, or camera localization. Such misleading claims do not help readers learn from the paper w.r.t related work in the community.


2) The approach "is rooted in the assertion that any deep CNN for image classification must contain, implicit in its connection weights, knowledge about the location of recognized object". This assertion does not appear obvious -- an reference should be cited if it is from other work. Otherwise, recent work shows that deep CNN can overfit random training data, in which case it is hard to imagine why the object location can be implicitly captured by the CNN [R1]. Similarly, the paper claims that "once weights are found, the gradient... with regard to X would provide information about the sensitivity of the bounding box loss function with regard to the pixels in the images". This is not obvoius either as recent work show that, rather than the whole object, a part of it may be more discriminative and captured by the network. So at this point, why the gradient can be used for object location without worrying that the model merely captures a part?

[R1] Chiyuan Zhang, Samy Bengio, Moritz Hardt, Benjamin Recht, Oriol Vinyals, Understanding deep learning requires rethinking generalization, ICLR 2017.
[R2] Bolei Zhou, Aditya Khosla, Agata Lapedriza, Aude Oliva, Antonio Torralba, Learning deep features for discriminative localization, CVPR 2016.

3) The paper admits in Section 2.2 that "we have not yet done a formal comparison of these two approaches to constructing the sensitivity map". As the two approaches are suggested by the authors, why not comparing in this paper. It makes the paper less self-contained and not ready to publish. A formal comparison in the rebuttal may improve the rating of the paper.

4) In Equation 3, how to represent the bounding box coordinate? Are they any transforms? What does it mean by "bias weights"? Are they different from Cartesian coordinates, or the one used in Equation (2)?


5) The experiments are not convincing by merely reporting the metric of IoU>0.5 without any in-depth analysis. Perhaps some visualization and ablation study improve the quality of experiments.

6) In Section 3.2, why using two different aggregation methods for producing the final sensitivity map -- max-pool along the channel for PACAL VOC 2017 dataset and avg-pool for ImageNet dataset, respectively? Are there some considerations?

7) In Table 1, it shows the proposed method outperforms the other methods significantly, achieving 41% better than the second best method. However, there is no in-depth analysis explaining why the proposed method performs so well for this task. Moreover, from Figure 1 and Figure 3, it is straightforward to ask how a saliency detection model performs in object detection given that the images have clean background and objects are mostly centered in the image.

8) What does it mean by "CorLoc (mAP)" in Table 2? As defined in Equation 4, CorLoc acounts the portion of detection whose IoU greater than 0.5 compared to the ground-truth. But mAP accumulates over a range of IoU threshold and precision across classes.

9) As the proposed method is closely related to the CAM method, how does CAM perform on these datasets? This misses an important comparison in the paper.


10) The readability of the paper should be improve. There are many typos, for example --
1. What does "..." mean above and below Equation (2)?
2. inconsistent notation, like $w_{ki}$ and ${\bf w}_{ki}$ in Equation (2).
3. conflicted notation, w used in Equation 2 and Equation 3.

---

### Official Review · AnonReviewer1 · 2018-10-30
**More oriented towards object localization rather than model interpretation**

**Rating:** 6
**Confidence:** 5

**Review:**


===================
SUMMARY
===================

The paper proposes a method to extend the output of a network trained for visual object recognition (i.e. image classification) with bounding box coordinates that can serve to localize the recognized object.
This process is referred to as "object localization", and it resembles to some extent the object detection task.
Object localization is achieved by analyzing the absolute values of the gradients in the last convolutional layer (referred to as "attention map" in the paper) with regard to the pixel locations of the input image.
Then, a linear model is trained in order to predict the bounding box coordinates that serve to localize the recognized object instance. This is different from traditional object detection methods, which learn how to predict bounding boxes directly from image pixels. The paper claims that by learning how to predict bounding boxes from sensitivity maps the amount of needed training data is reduced.

Experiments on the PASCAL VOC'07 and a subset of the ImageNet'12 dataset show the performance of the proposed method.

===================
REVIEW
===================

The content of the paper is clear, has a good flow. Moreover, the proposed method is sound and relatively easy to follow. The reported experiments show good performance of the proposed method. In addition, there seems to be hints that the proposed method has a computation speed that makes it suitable for on-the-fly computations.

My main concerns with the manuscript are the following:

The manuscript proposes to analyze the internal spatial activations/gradients of a network trained for object recognition with the goal of exploiting internally encoded cues that can serve for object localization. At the high level, as recognized in the manuscript, this is very similar to the work from Zhou et al.,CVPR'16 with some differences regarding how the attention maps are defined, and if additional components are added to the network.
In addition, the way in which the proposed object localization problem is formulated bears resemblance to work on weakly-supervised object detection where object detectors are trained by using only image-level annotations (no bounding box annotations are used).
Keeping these two groups of work in mind, I find the contribution of the proposed method limited.
Perhaps a explicit positioning with respect to this work is required.

Section 2.3 states that there two possibilities, i.e. max or average. to aggregate the sensitivity maps obtained for each of the RGB channels from images. However, in Section 3.2, it is stated that the max operation is used for the Pascal VOC'07 dataset while the average operation was used for ImageNet data. This inconsistency in the way these maps are aggregated makes it hard to observe if there is a difference or trend across these two operations. Is there a reason why different operations were applied to different datasets? I recommend applying both operations on both datasets and discuss observations from the results.

While the proposed method have been evaluated on two datasets, and compared with respect to standard classic baselines, a deeper evaluation analyzing the inner workings of the method is not conducted.
In this regard, in addition to evaluating the effect of the method to aggregate sensitivity maps accross color channels (previously mentioned), I suggest reporting results by changing the threshold used in intersection over union (IoU) measure used to verify matching (Section 3,1). In addition, I am curious about the performance of the method in the two following scenarios: i) when multiple instances of the recognized object are present in the image, and ii) when instances of other distractor classes are present in the image.

In several parts of the manuscript, e.g. Sec. 1, 3.4, and 4, claims regarding the speed of the proposed method are made. However, there is no evaluation focusing on this aspect that could be used to support this claims. In this regard, either a evaluation focusing on the computation speed of the proposed method should be conducted or claims regarding to computation speed should be toned down.

Finally, the first title of the manuscript, i.e. "Interpreting Deep Neural Networks", suggests that the manuscript will cover topics regarding model interpretation of DNNs. However, model interpretation as such is never touched wich makes this title somewhat misleading. In fact the main theme of the manuscript is about object localization/detection, hence the second part of its title: "Fast Object Localization via Sensitivity Analysis".
If this manuscript is to be accepted, the first part of the title should be removed.

I would appreciate if my main concerns are addressed in the rebuttal.

---

### Official Review · AnonReviewer2 · 2018-11-02
**Reasonable but more evaluation needed**

**Rating:** 4
**Confidence:** 3

**Review:**

Summary
The paper presents a method to perform object localization by computing sensitivity of the network activations with respect to each pixel. The key idea is that the representation for classification implicitly contains object localization information since the object classification is done by detecting features of an object in an image. The localization information is extracted as a form of sensitivity map which indicates each pixel’s contribution to the final classification decision, and is subsequently used for regressing the bounding box. The proposed method outperforms other baseline methods and achieves reasonable performance when compared with slower state-of-the-art deep learning techniques for object localization.

Strengths
-	For object localization, this technique can provide faster results.
-	The paper proposes a simple sensitivity measure which works well in identifying the pixels which are important for object classification, and provides relevant information for localization.
-	The paper suggests a simple linear mapping from sensitivity maps to object bounding box coordinates, which can be learnt from a fairly small ground truth localization data.

Weaknesses
-	The idea of utilizing back-propagated sensitivity map is not novel for weakly-supervised object localization [1,2], as the proposed method just uses a simpler sensitivity function with linear regression.
-	The paper mentions the approach being ‘very fast’, but they do not show any quantitative comparisons in fair and similar settings with other techniques. The reviewer strongly suggests to provide testing time measures.
-	In ImageNet experiment, test data split in the paper is not held out as they use a pre-trained VGG network which is trained on the full ImageNet dataset.
-	The title claims to interpret deep neural networks, but the proposed approach just uses sensitivity map for object localization task, without any analysis for interpretation. Such interpretations have already been extensively studied in [1,2].
-	The idea of utilizing classification network for localizing an object is new, but the ideas of weakly-supervised object localization is already explored in [1,2,3,4,5,6]. The reviewer recommends to cite and provide valid comparison with [3,4,5,6].
-	More detailed experimental results, i.e. accuracy across different categories are required. The reviewer also recommends ablation studies to compare with bounding box heuristics and sensitivity measures as used in [1, 2].
-	No details about reproduction of results are provided in the paper.

Possible Questions
-	When computing a sensitivity map from a CNN output vector or an attention map, is the sensitivity calculated with respect to each activation in the output vector? How is a single sensitivity map computed from the attention map, which contains a number of activations?

Minor Comments
-	In the description of g’(\eta_i), g’(\eta i) should be g’(\eta_i).
-	“…” around equations should be replaced by “: equation,”.

References
[1] Bolei Zhou, Aditya Khosla, Agata Lapedriza, Aude Oliva, and Antonio Torralba. Learning deep features for discriminative localization. In CVPR, 2016.
[2] Ramprasaath R Selvaraju, Michael Cogswell, Abhishek Das, Ramakrishna Vedantam, Devi Parikh, and Dhruv Batra. Grad-CAM: Visual explanations from deep networks via gradient-based localization. See https://arxiv.org/abs/1610.02391 v3, 7(8), 2016.
[3]  Cho et al., Unsupervised Object Discovery and Localization in the Wild: Part-based Matching with Bottom-up Region Proposals, CVPR 2015
[4] Yi et al., Soft Proposal Networks for Weakly Supervised Object Localization, ICCV 2017
[5] Oquab et al., Is object localization for free? – Weakly-supervised learning with convolutional neural networks, CVPR 2015
[6] Li et al., Weakly Supervised Object Localization with Progressive Domain Adaptation, CVPR 2016

---

### Meta-Review · Area_Chair1 · 2018-12-17

**Confidence:** 4
**Recommendation:** Reject

**Metareview:**

This paper was reviewed by three experts. After the author response, R2 and R3 recommend rejecting this paper citing concerns of novelty and experimental evaluation. R1 assigns it a score of "6" but in comments agrees that the manuscript is not ready for ICLR. The AC finds no basis for accepting this paper in this state.